# Indirect Dispersion of SARS-CoV-2 Live-Attenuated Vaccine and Its Contribution to Herd Immunity

**DOI:** 10.3390/vaccines11030655

**Published:** 2023-03-14

**Authors:** Ursino Pacheco-García, Jeanet Serafín-López

**Affiliations:** 1Department of Cardio-Renal Pathophysiology, Instituto Nacional de Cardiología “Ignacio Chávez”, Mexico City 14080, Mexico; 2Department of Immunology, Escuela Nacional de Ciencias Biológicas (ENCB), Instituto Politécnico Nacional (IPN), Mexico City 11340, Mexico

**Keywords:** LAVs, SARS-CoV-2 attenuated virus, herd immunity, indirect dispersion, COVID-19

## Abstract

It has been 34 months since the beginning of the SARS-CoV-2 coronavirus pandemic, which causes the COVID-19 disease. In several countries, immunization has reached a proportion near what is required to reach herd immunity. Nevertheless, infections and re-infections have been observed even in vaccinated persons. That is because protection conferred by vaccines is not entirely effective against new virus variants. It is unknown how often booster vaccines will be necessary to maintain a good level of protective immunity. Furthermore, many individuals refuse vaccination, and in developing countries, a large proportion of the population has not yet been vaccinated. Some live-attenuated vaccines against SARS-CoV-2 are being developed. Here, we analyze the indirect dispersion of a live-attenuated virus from vaccinated individuals to their contacts and the contribution that this phenomenon could have to reaching Herd Immunity.

## 1. Introduction

In the face of the appearance in China in December 2019 of a new human infectious disease called COVID-19, caused by the SARS-CoV-2 coronavirus [1,2,3,4,5], countries implemented governmental measures to control the virus dispersion among the population. The virus showed a high infection frequency in humans [6,7], and from the beginning, there was great difficulty in containing its spread through confinement [8]. Moreover, there was a high frequency of severe and lethal cases in elderly persons, mainly those with comorbidities such as diabetes and hypertension [9]. SARS-CoV-2 coronavirus propagated to several countries in February and March 2020 [10,11,12,13], and in the same year, the WHO declared a pandemic on March 11 [14].

At the start of the SARS-CoV-2 world dispersion, some governmental leaders (in countries from northern Europe, for example) proposed that its propagation could be controlled by allowing the free infection of individuals, which would induce protection by antibodies and cellular immunity in a high proportion of the population, thus achieving herd immunity (HI). That would lead to the consequent reduction or even elimination of the infectious agent [15]. Very soon, the proposition was strongly challenged [16,17,18] because achieving collective immunity through the infection dispersion could have a high cost on human lives and health complications caused by COVID-19 in elderly persons, mainly in those with comorbidities [1,9]. It was observed that although the percentage of lethality in the population was low in general, the high incidence of infection raised the absolute number of severe and lethal cases [19,20,21] so, as a better option, it was decided to control propagation through the containment of human activity to reduce the contact between individuals. Social mobility was restricted to allow only the circulation of persons dedicated to essential activities [20,21,22,23], waiting for the identification of effective antiviral drugs to treat infected individuals and the approval of efficient vaccines to achieve HI through massive vaccination [24]. In several countries, quarantine was strictly enforced, attaining a significant decrease in the number of cases during the first wave of infections [24,25,26]. Nevertheless, a few months after the pandemic’s beginning and with a still high incidence of infections and high numbers of deaths, the gradual return to essential and non-essential human activities was allowed in most countries. The decision was taken due to the population’s demand to restart their economic activities, which would be further affected by a quarantine extension [27,28,29,30,31,32]. After human activities were restarted, contagion increased again in several countries, with several waves of cases at different times and places [31,32]. For this reason, in the following months, there was a partial tolerance for the realization of economic activities, combined with partial social distancing, personal hygiene and other protection measures [33,34]. In December 2020, the massive application of different types of non-proliferative vaccines started in several countries [35,36]. Although in many developed and some developing countries, several booster shots have been administered, there are underdeveloped countries where vaccination has been delayed or is still very limited, mainly due to economic limitations, which prevented them from accessing vaccines since the first days after their approval [37,38,39,40,41]. There are still many developing countries where a sufficient proportion of the protected population has not been reached to be near the herd immunity threshold (HIT) against SARS-CoV-2 [39,40,41]. The risk of massive infections in several countries is latent, as well as the risk of the appearance and propagation of more dangerous and contagious mutant strains if the virus is still propagating in populations with low immunization rates [41]. On the other hand, in several countries, a proportion of the population opposes vaccination [42,43], which hampers the achievement of the herd immunity threshold, in addition to the fact that in many developing and underdeveloped countries, child vaccination is still low or inexistent [44,45].

After the massive application of vaccines, severe cases and mortality decreased sharply, as well as recovery time for the anew infected and the re-infected. Although there is a high proportion of immunized persons, the individual neutralizing antibody levels decline after some time, so epidemic outbreaks keep appearing due to new virus variants, some of which have been dispersed globally and others more locally. Nevertheless, as vaccinated individuals and those that recovered from infection maintain a certain degree of immunity, if they are re-infected, their symptoms are less severe, and they show lower mortality [46,47,48]. All of the above make necessary the application of boosters to help maintain protective immunity levels in the population in the face of the dispersion of new variants [49,50,51].

Here we address the topic of how a massive application of LAVs vaccines could result in the indirect dispersion of attenuated virus between non-vaccinated persons, which could help achieve the herd immunity threshold. In addition, it could help to maintain, for a longer time, optimal levels of humoral and cellular immunity in directly vaccinated individuals as well as in those that get the attenuated virus indirectly. The proportion of individuals indirectly immunized in this way might be considered for the theoretical calculations of viral dispersion and the achievement of HI.

## 2. Population Immunity

Individual immunity against a pathogen is a state in which the different components of the immunological system are prepared to protect her/him from this microorganism by controlling or eliminating it in case of infection [52,53,54]. In a given population, there is a certain proportion of individuals with immunity against one particular pathogen. If a pathogen is new for a population, the proportion of immune individuals is probably zero, and this population is likely to be highly susceptible to becoming infected by this new pathogen. Herd immunity is a population condition where the pathogen dispersion between the community members is difficult or impossible because when an infected individual appears, the individuals surrounding her/him are immune against the pathogen, so they do not get infected and do not transmit the pathogen to other susceptible individuals. Besides, in a population with a high proportion of immune individuals, the probability of an encounter between an infected and a susceptible individual is very low [55,56,57,58,59,60,61,62]. Herd immunity threshold (HIT) refers to the fraction of the population required to be immune against an infectious pathogen to prevent its dispersion. A population has reached herd immunity when it has a proportion of immune individuals against a particular pathogen equal to or above the HIT [62,63]. To calculate the proportion of the immune population required to reach the HIT, one must consider the pathogen dispersion capacity, which is given by the reproduction number R_0_, which indicates the average number of non-immune individuals whom a sick individual infects [55,56,57,58,59,60,61,62,63,64,65,66,67,68]. The fraction of the population required to reach HI in populations with a homogeneous immune response elicited by highly effective vaccines is calculated by the formula 1 − 1/R_0_. Nevertheless, the HIT can be calculated more precisely by considering several population factors and vaccine effectiveness [52,60,64]. It is generally considered that 70% percent of immune individuals against a particular pathogen in a population confers HI [52,58,65,66,67,68,69,70,71,72,73,74]. A similar percentage has been considered necessary for the SARS-CoV-2 case considering an R_0_ = 3 and a HIT = 67 [58,65,66,67,68,69,70,71,72,73]. This percentage might vary between countries or regions [67,74,75]. Immunization against SARS-CoV-2 reduces susceptibility but does not totally protect against infection or reinfection, so it is estimated that the required HIT might be higher, reaching 95% for some populations [52,59,60,63,64,66,67,71,73,74].

A population acquires immunity and reaches the HIT in three ways: (a) through the contagion of individuals with the wild-type strain. The individuals develop the disease with different symptomatology degrees and develop protective immunity, (b) through vaccination with different vaccine types, including inactivated vaccines, non-proliferative viral vector vaccines, and live-attenuated vaccines. (c) through the combination of natural dispersion and massive vaccination [75]. Here, we analyze an additional mechanism that might contribute to the development of collective immunity: (d) through the dispersion of viruses from live attenuated virus (LAVs) vaccines from vaccinated to non-vaccinated individuals. With the wide use of these LAVs, these viruses could disseminate in the same way as the wild-type virus causing mild symptoms while inducing an effective immunity against the pathogenic virus [76,77,78,79].

## 3. Immune Response against SARS-CoV-2 Induced by the Infection

Individual immunity is due to the development of antigen-specific antibodies and immune cells against the pathogen after surviving or clearing infection or after vaccination [80,81,82]. An appropriate level of humoral and cellular immunity prevents the same pathogen from infecting the immune individual, hindering its dissemination in the population [80,81,82]. Depending on the immunological status of the host and the viral load, her/him develops different antibody and cellular immunity levels that effectively protects her/him for a certain time. In some cases, individual immunity induced by infection can be sterilizing and lifelong, as is the case for smallpox [80,81,82,83,84].

For SARS-CoV-2, infected individuals develop humoral and cellular immunity against a large variety of its antigens. Neutralizing antibodies are directed against the S1 protein, which is the cellular counter-receptor that binds to the ACE2 protein on human cells to initiate the infection [85,86]. Anti-S1 antibodies prevent the infection of ACE2+ cells and promote the phagocytosis of free viruses and their destruction by phagocytic cells. Antibodies against other SARS-CoV-2 proteins are also developed, mainly against M, E, and N antigens [85,87]. Antigen-specific cytotoxic T-cells recognize infected cells through their specific T-cell receptors and eliminate them by cytotoxicity [88,89]. Immunity developed by infection is both systemic [85,86,87,88,89,90,91,92,93,94,95,96,97,98,99,100] and mucosal, mainly mediated by IgA antibodies [90,91,92,93,94,95,96,97,98,99,100].

## 4. Immune Response against SARS-CoV-2 Induced by Vaccination

During the COVID-19 pandemic’s first years, some of the given vaccines used the S1 protein as an antigen, while others used inactivated whole virus. Both induced mainly systemic immunity as they were administered intramuscularly (IM) [101,102,103,104,105,106,107,108,109,110,111,112]. Humoral and cellular immunity developed by the individuals that were immunized with S1 protein-based vaccines was directed against this antigen [102,103,104,105,106,107,108,109,110,111,112,113,114,115,116], whereas those that were inactivated virus-based, induced responses against the different viral antigens [116,117,118], including the S1 protein [117,118,119,120,121,122,123]. These same vaccines were modified for intranasal use and are under preclinical and clinical studies to establish if they can induce effective mucosal immunity [124,125,126,127,128].

Unfortunately, the immunity conferred by natural infection and the different vaccines is not sterilizing, or the level of protective immunity diminishes after some time. For that reason, infections or reinfections have been observed in both cases [129,130], although with less severe symptoms [131,132,133,134,135]. New virus varieties keep spreading among the populations, seriously affecting the unvaccinated or those with comorbidities [129,130,131,132,133,134,135], which is an indication that herd immunity has not been achieved in the different countries, regions, or population sectors [42,135,136,137,138,139,140,141,142]. LAVs vaccines could improve this situation as they induce a better immunological response without causing severe disease symptoms, and they stimulate innate and adaptive immunity, both systemic and mucosal. Some LAVs have contributed to controlling other viral diseases, such as smallpox, measles, or poliomyelitis, more effectively than molecular vaccines [143,144].

## 5. Advance in Obtaining Herd Immunity against SARS-CoV-2

During the first year of the pandemic, no vaccines were available, so in all countries, individual immunity was induced only through infection. In December 2020, massive vaccination started in some developed countries; bit by bit, other countries had access to vaccines. After that, herd immunity is attained in the three ways described by Lipsitch [75]:

(a) natural dispersion of SARS-CoV-2: to obtain individual and collective immunity, the easiest, quick, and most economical way is to allow the natural dispersion of the pathogen while massive vaccination must wait for licensed, effective, and safe vaccines [15,75,143]. In the first pandemic’s months, millions of people recovered from the infection and developed immunity. Around 90% of the infected showed mild or moderate symptoms. Unfortunately, 10% to 20% showed complications, and there was any effective antiviral drug to treat these cases. As a result, 2% to 3% of total cases became fatal. As total cases increased, the number of mortal victims became very high. A partial but functional immunity against SARS-CoV-2 is conferred to individuals previously in contact with other low pathogenicity coronavirus varieties with different homology degrees to SARS-CoV-2 [75]. If this immunity against other less pathogenic coronavirus varieties could be extended among the populations, this could help maintain an acceptable individual immunity level in most populations, and HI could be achieved with less difficulty [75];

(b) SARS-CoV-2 vaccine application: less than a year after the pandemic, several vaccines became available and started to be used massively. Some are based on the S1 protein, which is carried by other viral vectors (ChAdOx1, Ad3 and Ad5, AZD1222, SPUTNIK V, Ad5-nCoV) [114,115]; others are based on mRNA that codifies for the S1 protein (BNT162b2, mRNA1273) [116,117,118]. Others use the inactivated whole virus [119,120,121] or the isolated S1 protein [122,123,124], or virus-like particles [144,145,146,147,148,149]. These non-proliferating vaccines were approved and licensed relatively quickly, and booster doses have been administered in several countries;

(c) combination of natural dispersion and massive vaccination: in some countries, the combination of natural infections and massive vaccination has resulted in a percentage of immune persons near the one required to achieve HI, even though in several of these countries, children have not been vaccinated [150,151] nor the individuals that oppose vaccination [136,137,138,139,140,141,142]. SARS-CoV-2 LAVs are still in preclinical and clinical studies, and it takes a longer time for their approval because more strict biosecurity criteria must be met [152,153,154,155,156,157]. 

## 6. Cellular and Humoral Immune Responses Induced by SARS-CoV-2 Live-Attenuated Vaccines

Vaccination with LAVs has shown to be more effective than vaccination with molecular vaccines. Its application causes mild disease symptoms while inducing an immune response similar to the one induced by the original pathogen. Vaccinated individuals develop a specific adaptive cellular and humoral immunity against a wide variety of viral antigens at the systemic and mucosal levels. Besides, due to its similarity with the wild virus, it also stimulates innate immunity [109,152,153,154,157,158,159,160,161,162,163,164,165,166]. Immunity induced by LAVs could allow the control of infections with new variants, avoiding severe infection symptoms and reducing the time of viral dispersal [109,149,167,168], leading to the reduction of new cases. Of the non-proliferating vaccines applied during the first two years of the pandemic, some of them induce systemic immunity only against the S1 protein, while those based on inactivated viruses induce a response to a wide array of antigens. However, in both cases, immunity, although systemic, decays after some time and induces a low mucosal immunity, allowing a certain degree of infections and re-infections [109,169].

## 7. Features of LAVs against SARS-CoV-2

LAVs vaccines are more effective than inert vaccines, because, like the wild virus, they can replicate inside the cells of the vaccinated individual, resulting in better systemic and mucosal innate and adaptive immune responses [170,171]. Usually, only one dose is necessary, although, in the case of SARS-CoV-2, the vaccination scheme with LAVs is still under evaluation and could be applied annually. In addition, to immunize against the new variants of SARS-CoV-2, attenuated viruses could be generated from already approved LAVs with the appropriate S protein gene inserted from the new mutant strain to generate updated LAVs that express the S protein mutations, which is feasible, as is commented by Yoshida et al. about a LAVs platform developed by them [154].

The development and application of these vaccines must be carefully monitored as the attenuated viruses can revert to pathogenic ones that might cause symptomatic problems at the application site, usually the nasal cavity for SARS-CoV-2 [171,172]. LAVs should not be applied to individuals who might be highly susceptible to viral infections, such as those with a genetic or acquired immunodeficiency, those receiving immunosuppression treatment after a transplant, or receiving anti-proliferative medication or radiotherapy, or individuals infected with the human immunodeficiency virus (HIV) [173,174]. It is also recommended that these individuals should not be in contact with LAVs vaccinated individuals as they might acquire the attenuated virus, which could proliferate unchecked under a depressed immune system causing severe health problems [173,174]. Other susceptible groups are the elder and those prone to developing “Long COVID”. Older adults (75 years old or older) should be immunized with non-proliferative vaccines to develop immunity, preferably mucosal immunity, against SARS-CoV-2, before the application of LAVs vaccines in the community. The elders suffering from one or more comorbidities should take measures to avoid contact with LAVs and with persons recently vaccinated with LAVs. Before these vaccines are licensed, it must be determined for how many days the vaccinated with LAVs can spread the attenuated virus to establish for how long they should avoid contact with persons from the susceptible groups, including the elders. Long COVID is generally present in persons with an immunological dysregulation, which makes them unable to control and eliminate the viral infection efficiently. Individuals with obesity, type 2 diabetes, hypertension, above 50–60 years old, and who are malnourished, among other stressful situations, are the most susceptible to developing it. They must be immunized with non-proliferative vaccines and receive advice and treatment to correct the pathologies that make them more susceptible, such as exercise, weight loss, hypoglycemic treatment, a balanced diet, and correction of vitamin deficiencies, among others. These interventions would contribute to improving their immunological status. Another option is to treat those that develop Long COVID caused by an attenuated virus from the LAVs with antivirals effective against SARS-CoV-2, such as redemsivir.

Some LAVs against SARS-CoV-2 are under preclinical and clinical essays [172,173,174,175,176]. These vaccines have the advantage that their storage and distribution require only refrigerator temperatures (02 to 08 °C) [177].

## 8. Indirect Dispersion of Live-Attenuated Virus Vaccines

The transmission of the live attenuated virus from LAVs vaccinated persons to close unvaccinated individuals induce in them a protective immune response. This event was observed during the use of live attenuated vaccines against poliomyelitis in children in the middle of the XX century. Children who received the live attenuated vaccine virus dispersed the virus to their schoolmates (institutional dispersion) and to their contacts at home (familial dispersion) that had not been vaccinated [178,179]. Several studies on this matter were carried out in the URSS and in the United States of America [179,180,181,182,183,184,185]. The application of this vaccine allowed to contain the pathogenic virus dispersal in a much more significant proportion than with the inactivated virus vaccine [179,180,181,184,185,186]. Studies were carried out in closed human communities to analyze the indirect dispersion of attenuated vaccine viruses. It was found that the propagation is between around a 10% to 30% [179,181,184,185,186,187], which, although variable, contributes to the increment of HI as it helps to immunize unvaccinated persons and might also help to increase protective immunity in persons with a single dose of vaccine, working in this case as a booster vaccine [187].

It is essential to maintain two key characteristics of the virus used for LAVs in the individuals that acquired them indirectly: their attenuation and their immunogenicity [165,187]. To be sure that the attenuated virus does not recover its pathogenicity, German Todorov and Vladimir N. Uversky propose that varieties of the SARS-CoV-2 virus that induced mild symptomatology must be attenuated by multiple passages [188] or attenuated strains could be developed through genetic engineering [152,153,168,172,175,187,189]. There are several techniques to generate attenuated viruses, such as the insertion of high replication genes and genes that prevent excessive mutations [157,178,179,180]. A good strategy to facilitate the indirect dispersion of the SARS-CoV-2 LAVs between individuals would be to seek that the attenuated virus maintains a degree of replication and contagion similar to the pathogenic virus in order to increment its dissemination.

The indirect propagation of attenuated virus from these vaccines is mainly through the respiratory tract, as is the case of the wild virus, so its degree of contagion could be similar to the pathogenic SARS-CoV-2 variants [188,189,190,191].

Several individuals infected with the SARS-CoV-2 virus are asymptomatic or show mild symptoms. Although the mild symptoms are attributed to a robust immune system, it is also possible that some of these individuals were infected with a low pathogenicity virus variant, and so could be the source of virus strains that already have a certain degree of attenuation. These strains could be a good starting point for developing an attenuated virus useful for a vaccine [173,174,175,176,177,178]. 

Some vaccines against SARS-CoV-2, currently under development, use as vectors of the S1 protein, an attenuated virus that protects against other diseases. Among them are vaccines against measles, influenza, modified vaccinia Ankara (MVA), and yellow fever [191,192,193,194,195,196,197,198,199,200,201,202,203,204,205]. Nevertheless, these vaccines use vectors that are of low dispersion, so their contribution to the achievement of herd immunity against SARS-CoV-2 is very low or non-existent. Rhinoviruses are highly infectious in humans and cause the common cold with mild symptoms in most of the infected individuals. Their low pathogenicity and high contagion rate could make them suitable vectors for viral vaccines, such as for SARS-CoV-2 [206]. Another possibility is to develop LAVs vaccines based on a seasonal cold coronavirus with high homology to SARS-CoV-2. Immunogenic sequences of the SARS-CoV-2 S1 protein could be inserted into this virus. Theoretically, as for the case of attenuated SARS-CoV-2, it would be more effective than the non-proliferative vaccines used during the first two years of the pandemic and probably better than vaccines based on vectors that immunize only against one or two SARS-CoV-2 antigens. Indirect dispersal of LAVs is another way that could contribute to achieving HI against SARS-CoV-2 (Table 1; Figure 1). Studies of the indirect dispersion of LAVs from vaccines can be carry-out in animal models [207,208,209,210,211,212,213,214] and could give an idea of how this indirect dispersion would behave in human populations [209,210,211,212,213,214,215,216].

## 9. Immunity Induced by the Indirect Acquisition of Attenuated SARS-CoV-2 Virus

Immunity induced by the indirect acquisition of live attenuated poliomyelitis virus is similar to the one induced by vaccination, both in non-immune individuals and in individuals already immunized [186,187,217], and a similar situation could be considered for the case of SARS-CoV-2. LAVs vaccines against SARS-CoV-2 in preclinical phases are given by the nasal or oral route. The virus replicates in the tissues of these places inducing immunity against all the attenuated viral antigens and is potentially retransmitted to other individuals by the airborne transmission of aerosol or saliva particles. In case a SARS-CoV-2 LAVs vaccine is approved for human use, the process of indirectly acquiring the LAV would not be risky as the vaccine should comply with all the biosafety requirements to be approved. The degree of immunity it would induce would depend on the capacity of immune response of each individual, the replication rate of the virus, as well as the initial virus load. According to the observations on the vaccination with LAVs against SARS-CoV-2, in the direct recipients of the vaccine, it stimulates the components of innate immunity. Adaptive mucosal and systemic immunity is also developed against all the attenuated virus antigens, and this immunity confers resistance against the SARS-CoV-2 infection [159,160,161,162,164,165,166,167,168,169].

## 10. Effect of the LAV’s Indirect Dispersion on Individual and Population Immunity

A LAV-based vaccine against SARS-CoV-2, considering the great proportion of the directly and the indirectly vaccinated, could help to maintain an efficient immune response in the individuals for several months, would help to control the viral dispersion or keep it at very low levels. That, and an aggressive combination of a booster vaccination with the existing vaccines and antiviral treatment for individuals with high viral load, might contribute to the virus’ eradication. Besides, a society culture on how to help achieve HI must be promoted to help contain and eradicate this infectious disease [218,219]. The application of LAVs with a certain frequency, for example, yearly, could be a way to maintain a high level of individual immunity enough to keep reinfections at bay. In addition, each vaccination campaign would allow the indirect dispersion of the LAV, increasing the proportion of immune individuals and favoring the HI (Figure 2).

So far, the different mathematical models to calculate the number of individuals required to reach HI do not include the proportion of the population that would be inadvertently immunized through the indirect spread of LAVs. Its inclusion in the predictive models would contribute statistical data that might be useful if these vaccines are used massively. Several aspects about the use of LAVs must be studied: the percentage of population that is indirectly immunized, propagation of the virus in different age groups, its propagation between previously vaccinated individuals with other anti-SARS-CoV-2 vaccines, duration of the attenuated virus infection, attenuation conservation, LAVs mutations while they are spreading in the population and the kind of immunity induced in the indirect recipients; to name some.

## 11. Conclusions

SARS-CoV-2 vaccines based on LAVs could induce better innate and adaptive immunity in the same way as other LAV-based vaccines against other pathogenic viruses. The indirect dispersion of attenuated virus could help to increase the collective immunity by allowing the infection of individuals in contact with vaccinated individuals who would develop an immunity similar to those that were directly vaccinated. The development of high contagiousness attenuated virus vaccines against SARS-CoV-2 could significantly increase the proportion of the immunized population through the indirect immunization of individuals who do not receive any kind of vaccines for any reason.

There are already methodologies to avoid the appearance of viruses that recover their pathogenicity, so it is possible to develop attenuated immunogenic viruses with this characteristic. In the face of the persistence of natural infections with different variants of SARS-CoV-2, even in immunized individuals, due to a gradual decline in the protective immunity and the variability of immune response between different groups of individuals, it is necessary to keep a high immunity level through frequent and massive vaccination, trying to reach a 100% vaccination ideally. We think that the application of LAVs vaccines against SARS-CoV-2 with a certain level of contagiousness should be considered once they comply with the efficacy and biosafety requirements to be used in humans. The use of LAVs vaccines and their indirect dispersion could help face other pathogens in future pandemics.

## Figures and Tables

**Figure 1 vaccines-11-00655-f001:**
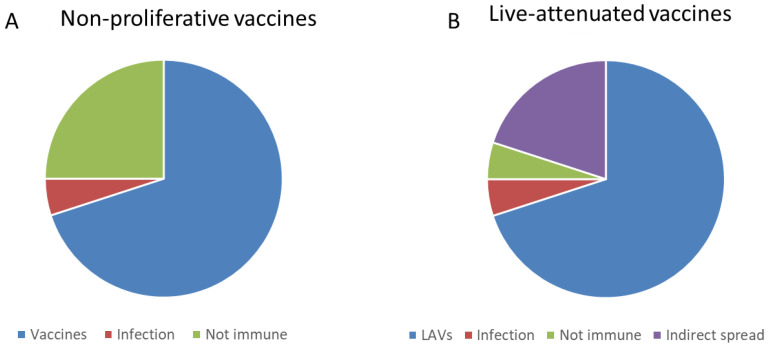
Proportions of immune individuals achieved by dispersal of the pathogenic microorganism and non-mandatory immunization with non-proliferative vaccines (**A**) compared to proportions achieved by non-mandatory application of proliferating LAVs (**B**). The transmission of the live attenuated virus from LAVs vaccinated persons to close unvaccinated individuals induces in them a protective immune response increasing the proportion of immune individuals favoring HI.

**Figure 2 vaccines-11-00655-f002:**
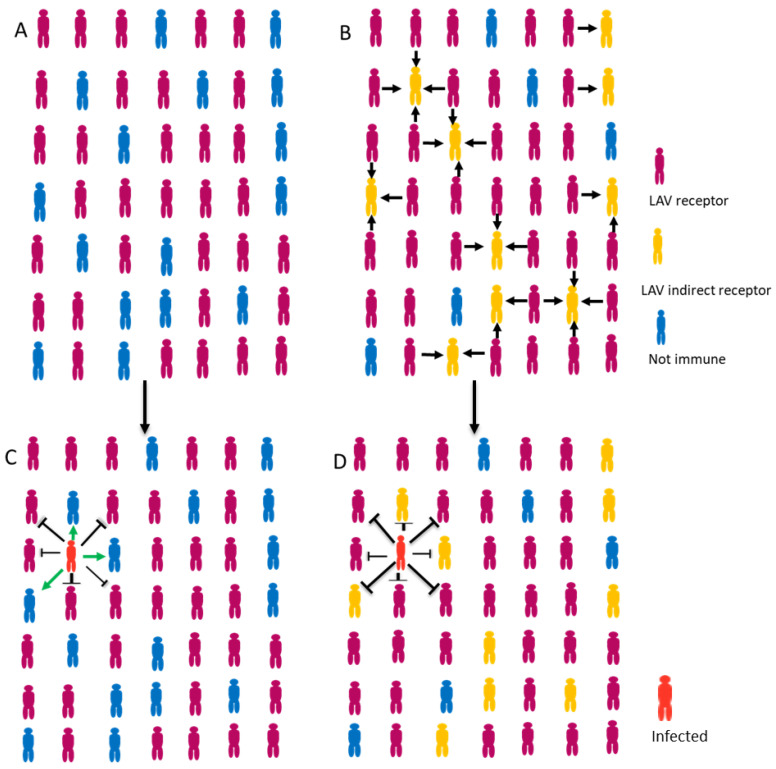
Schematic representation of the HI achieved with the non-proliferative vaccine application (**A**) compared with the hypothetical HI achieved with LAV application (**B**). Non-proliferative vaccines induce immunity in individuals who receive them; LAV vaccines induce immunity in individuals who receive them, as well as those contacts in whom the attenuated virus spreads, increasing the % of the immune population. The spread of the pathogenic SARS-CoV-2 in a population with 60–70% immune individuals is still possible (**C**), but it would be more difficult or would no longer occur in a population with a higher of immune individuals (**D**).

**Table 1 vaccines-11-00655-t001:** Pathways that contribute to HI against the SARS-CoV-2 coronavirus.

Spread of SARS-CoV-2 between immunized and non-immunized individuals [75]. Immunization by direct vaccination with any type of vaccine [75]. Exposure to other coronaviruses with some degree of homology to SARS-CoV-2 [75]. Indirect exposure to attenuated SARS-CoV-2 viruses by contact with individuals vaccinated with LAVs.

## Data Availability

Data sharing is not applicable to this article.

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
