# Peer review of "Indirect Dispersion of SARS-CoV-2 Live-Attenuated Vaccine and Its Contribution to Herd Immunity"

_vaccines, 2023, doi:10.3390/vaccines11030655_

Round 1

Reviewer 1 Report

A comprehensive review of LAVs vaccines against SARS-CoV-2 is likely to be unique. Unlike mRNA vaccines, LAVs vaccines have the characteristic of being able to transmit the attenuated virus from LAVs vaccinated persons to nearby unvaccinated individuals. This indirect dispersion should contribute to herd immunity more efficiently than mRNA vaccines. However, there are still many unknowns about SARS-CoV-2, and even attenuated, it is quite risky to use the virus as a vaccine. Therefore, it must be said that its feasibility is extremely low at this stage. The contents of this review are of considerable interest and the manuscript is well-written. However, I think it is necessary in this review to discuss not only the advantages of LAVs vaccines, but also weak points as described in the specific comments, although some has already been written about the possibility of attenuated viruses reverting to pathogenic viruses (such as Lines 228-238; 258-269; 279-296). 

Specific comments:

1)     Even if the attenuated SARS-CoV-2 does not revert to a virulent one, people who have been vaccinated with the LAV vaccine or who have not been vaccinated but have been infected with the attenuated virus through vaccinated persons may suffer from Long COVID. How do the authors think about this issue?

2)     Those who become seriously ill or die from COVID-19 are generally aged persons, as well as those with weakened immune systems (Lines 230-234). How are the authors planning to devise and use the LAVs vaccine for the elderly?

3)     It is well known that mRNA vaccines are not fully effective against mutant viruses. Furthermore, the immunity induced by mRNA vaccines does not last long, and hence infections with SARS-CoV-2 are observed even in vaccinated persons. I am sure the LAVs vaccine will behave quite similarly. How do the authors deal with this issue?

Reviewer 2 Report

Manuscript untitled „Indirect dispersion of SARS-CoV-2 live-attenuated vaccine and its contribution to herd immunity. ” is a nice review about different type of vaccinationns that we have against SARS-CoV-2 infection.

Article is divided into 11 parts and well organized. Authors starts with small introduction in which described COVID-19 as a diseases and described in details pandemic situation and stages of it. Authors also try to explan why they think LAV vaccines will bring better efficiency to anti-COVID 19 immune resistance in society.

In chapter 2 there are described the ways how we get the immunity. While in chapter 3 is presented Immune response against SARS-CoV-2 induced by the infection and in chapter 4 Immune response against SARS-CoV-2 induced by vaccination.

In part 5 authors emphasize advances in obtaining herd immunity against SARS-CoV-2 using different methods from infections and varies types of immunization.

In next chapers (from 6 to 10) authors trying to prove dominance of live attenuated vaccines against SARS-CoV2 in reaching hest immunity. They described cellular and humoral immune responses induced by SARS-CoV-2 and immunity induced by the indirect acquisition of attenuated SARS-CoV-2 virus and the main advantages of LAVs.

Authors concluded on the end that SARS-CoV-2 vaccines based on LAVs could induce better innate and adaptive immunity in the same way as other LAVs-based vaccines against other pathogenic virus and this kind of vaccination should be also in future take into consideration because make higher efficiency of vaccination of societies.

To sum up, I give a positive opinion about manuscript untitled „Indirect dispersion of SARS-CoV-2 live-attenuated vaccine and its contribution to herd immunity”.

Round 2

Reviewer 1 Report

I have no serious criticisms in the revised manuscript.